# Impact of a national primary care pay-for-performance scheme on ambulatory care sensitive hospital admissions: a small-area analysis in England

Christos Grigoroglou [1,2] Luke Munford [2,3] Roger Webb,[2,4,5] Navneet Kapur,[2,5,6,7] Tim Doran,[8] Darren Ashcroft,[1,2,5,9] Evangelos Kontopantelis [1,10]

For numbered affiliations see end of article.

**Correspondence to**
Dr Christos Grigoroglou;
christos.grigoroglou@postgrad.manchester.ac.uk

## ABSTRACT

**Objective** We aimed to spatially describe hospital admissions for ambulatory care sensitive conditions (ACSC) in England at small-area geographical level and assess whether recorded practice performance under one of the world's largest primary care pay-for-performance schemes led to reductions in these potentially avoidable hospitalisations for chronic conditions incentivised in the scheme.

**Setting** We obtained numbers of ACSC hospital admissions from the Hospital Episode Statistics database and information on recorded practice performance from the Quality and Outcomes Framework (QOF) administrative dataset for 2015/2016. We fitted three sets of negative binomial models to examine ecological associations between incentivised ACSC admissions, general practice performance, deprivation, urbanity and other sociodemographic characteristics.

**Results** Hospital admissions for QOF incentivised ACSCs varied within and between regions, with clusters of high numbers of hospital admissions for incentivised ACSCs identified across England. Our models indicated a very small effect of the QOF on reducing admissions for incentivised ACSCs (0.993, 95% CI 0.990 to 0.995), however, other factors, such as deprivation (1.021, 95% CI 1.020 to 1.021) and urbanicity (0.875, 95% CI 0.862 to 0.887), were far more important in explaining variations in admissions for ACSCs. People in deprived areas had a higher risk of being admitted in hospital for an incentivised ACSC condition.

**Conclusion** Spatial analysis based on routinely collected data can be used to identify areas with high rates of potentially avoidable hospital admissions, providing valuable information for targeting resources and evaluating public health interventions. Our findings suggest that the QOF had a very small effect on reducing avoidable hospitalisation for incentivised conditions. Material deprivation and urbanicity were the strongest predictors of the variation in ACSC rates for all QOF incentivised conditions across England.

## INTRODUCTION

Worldwide, unplanned hospital admissions are undesirable for patients, disruptive of

### Strengths and limitations of this study

► This study uses data from a rich dataset to investigate the association between recorded general practice performance and hospital admissions for Quality and Outcomes Framework (QOF) incentivised ambulatory care sensitive conditions (ACSCs) for the whole of England.

► Previous evidence suggests moderate effects of the QOF on admissions for incentivised ACSCs and our work shows that area characteristics account for more variation in admissions for incentivised conditions than QOF recorded performance.

► There is substantial spatial variation in admissions for incentivised ACSCs at a very low geographical level (ie, lower super output area).

► Hospital admissions for ACSCs are a commonly used measure of system performance, however, this is sometimes considered problematic and emphasis should be given to databases that have a greater focus on processes of care.

► Even though we cannot rule out the possibility of ecological fallacy our findings are important from a policy perspective due to the high number of potentially avoidable admissions and the associated high cost to the National Health Service.

elective care and costly.[1] As such, design and implementation of interventions targeting emergency admissions that are potentially avoidable has become a key priority for the efficiency and effectiveness of health systems. Ambulatory care sensitive conditions (ACSCs) are acute and chronic conditions for which hospital admissions can potentially be prevented through effective management,[2] primary prevention[3] and high-quality primary care.[4]

In England, it is estimated that ACSCs account for one in five unplanned admissions[5] and recent evidence shows that the number of emergency admissions for ACSCs

increased by 48% between 2001 and 2013.[5] Admissions for ACSCs are estimated to cost the National Health Service (NHS) £1.42 billion annually, equivalent to an average cost of £1739 per ACSC admission and an average cost of £170 590 per general practice per year.[6]

Efforts to improve the quality of primary care in the UK led to the introduction in 2004 of what was then the world's largest primary care pay-for-performance scheme. The UK's Quality and Outcomes Framework (QOF) was launched as part of a new national contract for general practitioners and in England continues to provide financial rewards to practices for meeting a range of performance indicators, primarily relating to the management of chronic conditions.[7] The scheme is essentially voluntary, but because the incentives substantially increase practice income around 96% of practices in England participated in the scheme in 2015/2016.[8] Evidence from the early years of the scheme suggests that the QOF reduced variations between practices in the delivery of incentivised processes of care,[9] and contributed to progress towards better use of electronic records and nurse-led multidisciplinary care of long-term conditions[10] but had no overall effect on mortality.[11 12]

The QOF was also associated with reduced hospital admissions for some chronic conditions[13–15] but no associations or mixed results have been reported for others.[16–18] In terms of ACSCs, many chronic conditions included in the QOF are potentially avoidable so, if the scheme is successful in improving quality for these conditions it should also lead to a reduction in hospital admissions for ACSCs.[19] Previous evidence on ACSC admissions suggested that the scheme may have reduced the trend in ACSC admissions for conditions that were included in the scheme relative to ACSC admissions that were not included in the scheme—by 3% in 2004/2005% and 8% in 2010—and this was mainly driven by relative reductions in emergency admissions for coronary heart disease.[19]

Recent national strategies have identified the need to quantify the extent to which high quality of primary care impacts on admissions for ACSCs as accumulating evidence indicates substantial unexplained geographical variation in ACSC admissions across England.[20] These variations have also revealed that hospital admissions for ACSC are concentrated in areas of high socioeconomic deprivation.[6] Furthermore, people living in deprived areas with greater health needs generally have poorer access to high-quality primary healthcare,[21] and this can have implications for the numbers of ACSCs admissions observed in England.

In this study, we aimed to: (1) spatially describe variations in hospital admissions for QOF incentivised ACSCs and QOF recorded performance for incentivised ACSCs in 2015/2016 at the small area level for the whole of England; (2) evaluate and quantify the relationship between QOF recorded general practice performance and hospital admissions for incentivised ACSC conditions and (3) examine whether deprivation and population characteristics such as age, sex, ethnicity and urbanity drive variability in this association across English regions.

## METHODS
### Data sources
We accessed various data sources to extract information on emergency hospital admissions for incentivised ACSCs, recorded quality of primary care, population size estimates, deprivation, disease prevalence and spatial coordinates. Data were collected at the lower super output area (LSOA) level, an administrative unit of English geography, which has a mean (and median) population size of around 1500 residents. We obtained data on all hospital admissions for incentivised ACSCs from the Hospital Episode Statistics (HES) national database from NHS Digital by year of presentation, broken down by gender and 5 year age bands, for the period 1 April 2006 to 31 March 2016. HES is a national database containing details of all admissions, emergency department (ED) attendances and outpatient appointments at all NHS hospitals in England. In HES, all admission data are related to episodes rather than persons as some individuals may have been admitted in hospitals multiple times over one calendar year and these presentations will be recorded as separate admissions. We considered only the first finished admitted episode in a spell of care as our aim was to focus on the reason of admissions rather than conditions which could develop later in the spell. Using International Classification of Diseases, 10th revision (ICD-10) codes for the primary diagnosis, we identified emergency admissions for ACSCs of the conditions which were incentivised under the QOF in our study period. This work focuses only on QOF incentivised ACSCs, not all ACSCs and we shall refer to these for the remaining of the manuscript as ACSCs. To measure ACSCs, we followed the work by Harrison *et al*.[19] Nine specific conditions were chosen based on the list of ACSCs which are used for measuring system performance in the NHS[22] and we also included complications of diabetes associated with hypoglycaemia.[14] The 9 QOF incentivised conditions included: asthma, coronary heart disease, chronic obstructive pulmonary disease (COPD), diabetes (separated into admissions for hyperglycaemia and hypoglycaemia), epilepsy, heart failure, hypertension and stroke. All relevant ICD-10 codes are provided in online supplementary table S1.

We derived data on population counts by gender and age group, ethnicity from the 2011 national Census. From the Office for National Statistics, we obtained data on LSOA rurality/urbanicity and local area deprivation based on the Index of Multiple Deprivation (IMD) 2015. The IMD measures relative levels of deprivation for all the 32 844 LSOAs in England on a continuous scale of deprivation where most of the indicators are based on 2012 statistics. In England, general practice healthcare coverage is comprehensive as 95% of the UK population is estimated to be registered with a practice, and over 99% of registered patients attend practices participating in the

QOF.[8] LSOA general practice performance data were obtained from the practice-level data in two ways. First, for those localities (ie, LSOAs) containing at least one general practice, practice-level information were aggregated at the LSOA level and this allowed us to account for multiple practices in an LSOA. These primary care practice 'hubs' varied between 6455 in the third year of the QOF (2006/2007) to 5736 in the year of our analysis (2015/2016). Second, for the remaining 80% of LSOAs that do not contain a general practice, we used spatial analysis techniques to estimate healthcare data at the LSOA level from the practice-level data.[12] These spatial analysis techniques are described in detail in the supplementary material for space reasons. All relevant information on number of practices, total list size and prevalence were available at the LSOA level. We captured QOF practice performance by calculating population achievement, defined as PAoval = $\frac{\sum N_i}{\sum (Di+Ei)}$[12] where the numerator represents the sum of all patients who have actually received the care (Ni) described in the relevant indicator (indicator $i$), the denominator represents the sum of the number of patients from the appropriate disease register who are eligible to receive the care described in the relevant QOF indicator ($Di$) and and the sum of the number of patients who are on the disease register but are not included in the relevant QOF indicator denominator for definitional reasons (exception reported) ($Ei$).[23 24] Population achievement was calculated within each financial year, across the full set of outcomes and measurement QOF indicators for incentivised conditions. All relevant QOF indicators with information on indicator name, years they were active, detailed description and their range are provided in online supplementary table S2.

## Statistical analysis

The primary outcome examined was hospital admissions for QOF incentivised ACSCs in 2015/2016. We used digital mapping software to visualise the spatial distribution of the outcome variable across England and within regions. We used negative binomial and zero inflated negative binomial models, which are suitable for analysing count data in the presence of overdispersion, and report incidence rate ratios (IRRs). We fitted one set of negative binomial and zero-inflated binomial models, weighted for 2015 LSOA population size, to quantify the association between admissions for incentivised ACSCs and quality of care, deprivation, demographic characteristics (age, sex and ethnicity), urbanicity and region (a description of the variables used is provided in the online supplementary material). As a sensitivity analysis, we used a second set of models to examine the association between admissions for incentivised ACSCs and quality of care, deprivation and demographic characteristics over time (ie, 2006–2015). A third set of analyses was performed with interaction terms fitted for region and deprivation score to assess whether the association between QOF population achievement and outcome varied across regions and

deprivation scores. A fourth set of analyses was performed with age-adjusted rates of admissions for incentivised ACSCs as the dependent variable with QOF population achievement, deprivation and rurality as the independent variables. Variation at the regional level was quantified through the third set of models, adjusting for other covariates. Stata V.15 was used for data management and all analyses.

## RESULTS

We present variability at the LSOA level for incidence of hospital admissions for incentivised ACSCs across regions in 2015/2016 in figure 1. There were 268 509 ACSC admissions with a primary diagnosis linked to QOF incentivised conditions in 2015/2016 alone while admissions for heart disease and COPD accounted for the majority of admissions for incentivised ACSCs over time (online supplementary figure S1). Per 100 000 people with incentivised ACSCs, the North East and North West regions had the highest overall admissions, whereas London had the lowest. For the whole of England, we present the spatial variability of admissions for ACSCs at the LSOA level in figure 2 where darker areas had higher ACSC admissions rates. The spatial variability of age-adjusted admissions for ACSCs is presented in online supplementary figure S2. We observed great variability for ACSC admissions across and within regions. We found large clusters of high numbers of admissions for ACSCs mostly in areas in the North of England and we generated regional spatial maps for our outcome of interest, provided in the online supplementary material. In figure 3, we present the spatial variability of quality of primary care across all conditions, for which we had data on ACSC admissions, at the LSOA level. We also present the spatial variability of admissions for incentivised ACSCs and quality of primary care for each English region at the LSOA level in the supplementary material (online supplementary figures S3:12 and S13:S22, respectively). Descriptive statistics on incentivised ACSC admissions, population size estimates, number of practices, census information, deprivation and QOF population achievement across regions for 2015/2016 are reported in table 1.

Median QOF population achievement across regions varied between 80.6% (London) and 84.3% (North East) in 2015/2016. At the LSOA level, population achievement varied from 33.3% to 91.47% in the third year of the QOF and from 57.01% to 98.25% in the tenth year of the QOF.

We present IRRs for the coefficients of interest in table 2. Results from our main model indicate that QOF population achievement led to small reductions in hospital admissions for incentivised ACSCs (IRR 0.993; 95% CI 0.990 to 0.995) where a 1% increase in quality of care nationally corresponds to a reduction of 187 admissions for all incentivised conditions in 2015/2016. This finding was consistent across all model specifications. For example, in the sensitivity analysis that investigated the

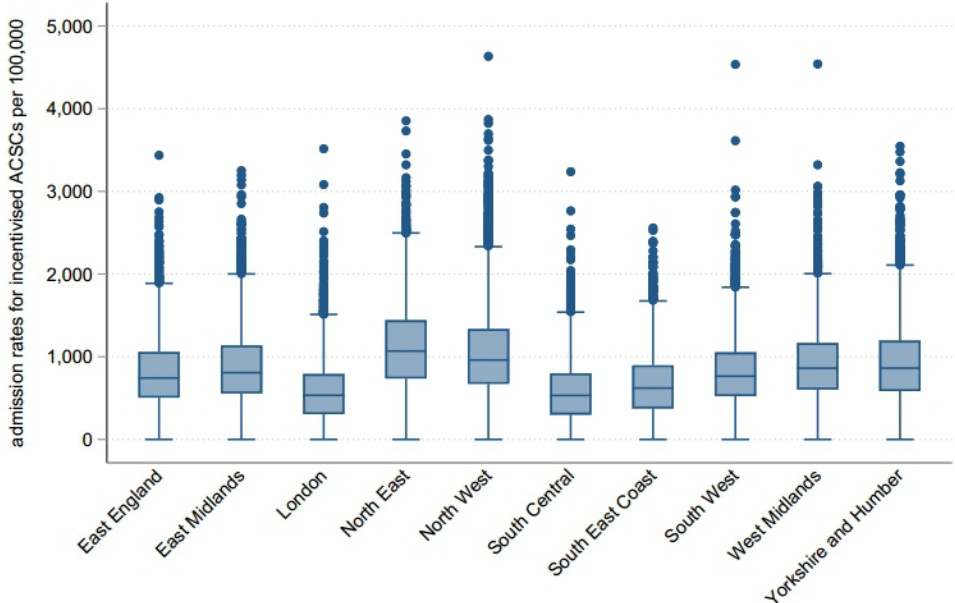

**Figure 1** Box plot of hospital admissions for QOF incentivised ACSCs, across English regions in 2015/2016 (weighted for 2015/2016 LSOA population size). ACSCs, ambulatory care sensitive conditions; LSOA, lower super output area; QOF, Quality and Outcomes Framework.

impact of QOF population achievement on admissions for incentivised ACSCs over time, results were nearly identical (IRR 0.998; 95% CI 0.997 to 0.999). Across all age groups, the highest risk for an ACSC admission was observed for the older groups and more markedly for the 85 years and older age group (IRR 18.448; 95 CI% 17.909 to 19.003). Female gender (IRR 0.852; 95% CI 0.845 to 0.860) and deprivation (IRR 1.021; 95% CI 1.020 1.021) were also strong predictors of ACSC admissions across all models. For example, inhabitants in rural areas had a lower risk of hospitalisation for QOF incentivised ACSC conditions (IRR 0.875; 95% CI 0.862 to 0.887). The results from the binomial regression were consistent across all models. All results from the negative binomial and zero inflated negative binomial models were identical (online supplementary table S3).

We also found evidence of heterogeneity in the association of interest by health region, indicating that in all regions higher QOF population achievement for incentivised conditions was associated with very small reductions in ACSC hospital admissions. Furthermore, our test for heterogeneity in QOF scores by deprivation indicated that the QOF resulted in slightly larger reductions in admissions for incentivised ACSCs in areas with higher deprivation scores. Results from both heterogeneity tests are presented in table 2. The result from the analyses using age-adjusted rates of admissions for incentivised ACSCs as the outcome variable revealed very small effects of QOF population achievement (online supplementary table S4).

## DISCUSSION
### Summary
This study provides evidence on the effectiveness of one of the world's largest primary care pay-for-performance schemes on avoidable hospitalisation for ACSCs. Our findings show that the QOF has had some impact on reducing avoidable admissions for incentivised conditions; however, the effect was very small and other factors such as deprivation and urbanicity more strongly predict risk of ACSC admission. We observed large variability in ACSC admissions across regions and we identified large clusters of high numbers of ACSC admissions within regions. This can help identify areas in greater need of interventions targeted at reducing potentially avoidable hospital admissions.

### Strengths and limitations
This study was conducted using routinely collected data on quality of primary care and hospital admissions for QOF incentivised ACSCs for the whole primary care-registered population of England. We generated population-wide maps of quality of primary care and hospital admissions for ACSCs and we examined the effects of deprivation at a small area level. This allowed us to account for demographic characteristics such as age and sex, which is not possible with analyses at the general practice level. Moreover, we visualised the parameters of interest and identified spatial clusters of high and low need for relatively homogeneous populations.

However, our study has several potential limitations which should be considered in interpreting the key findings. First, the possibility of ecological fallacy cannot be ruled out as we assigned practice level information to small-area localities and this did not allow us to determine how much of the ecological association is explained by variation in the distribution of individual-level risk factors. However, the lack of individual-level data makes the QOF the primary source of information

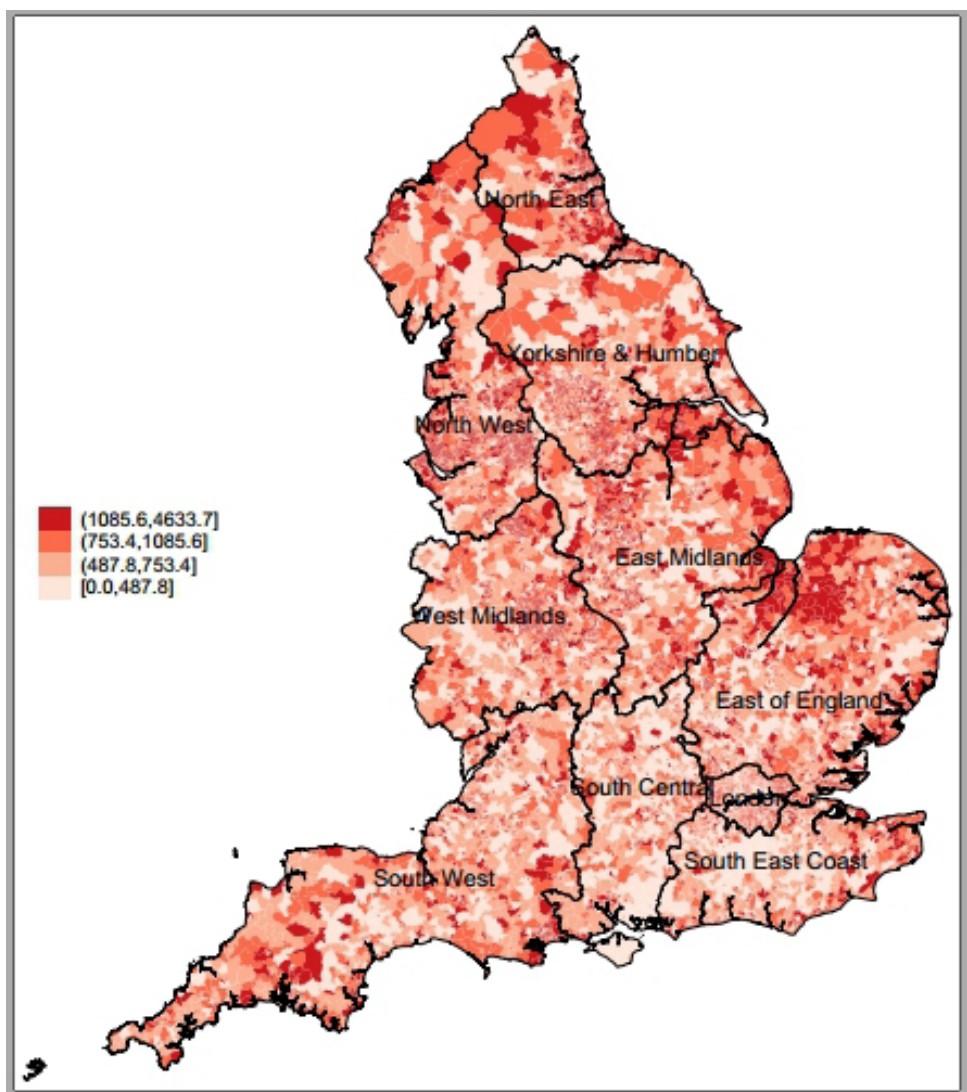

**Figure 2** QOF incentivised ACSC admissions per 100 000 population in England in 2015 at the LSOA level. ACSCs, ambulatory care sensitive conditions; LSOA, lower super output area; QOF, Quality and Outcomes Framework.

for quality of primary care in England, which enabled us to estimate data at a very low geographical level. An alternative approach would warrant the use of multilevel model methods, although this can only be tested if national individual-level data become available in the future. Second, administrative data, such as the HES database, lack nuanced information that may be essential in determining whether a hospitalisation could have been avoided (eg, severity of disease, social factors and patients' concerns and expectations) through high-quality primary care.[25] Third, ethnicity data were available through the census at the LSOA level as '% white' and we could not access age group-specific population measures for ethnic identity. Fourth, in the first 2 years of the scheme quality for most incentivised activities in the QOF improved at a faster rate and remained at high levels from the third year onwards, even though previous evidence argues that the rate of improvement has possibly occurred at the expense of other non-incentivised aspects of care.[26] We might, therefore, have failed to identify a larger reduction in

admissions for incentivised ACSCs in the first 2 years of the scheme. Fifth, we could not incorporate additional confounding factors such as continuity of care or access to care. These factors may influence the levels of incentivised ACSC admissions and future research should look to evaluate the effects of these factors should individual routine level data containing this information become available. Sixth, reliance on ACSCs as a measure of system performance is sometimes considered problematic and emphasis should be given to investment in alternative sources of data with a greater focus on processes of care (eg, clinical audits and registries). Until then, admissions for ACSCs are one of the best sources of information available to researchers and policy-makers to understand the complex relationships between patient outcomes and care received.

### Interpretation of findings
Hospital admissions for ACSCs have been used as a tool for monitoring healthcare systems in multiple countries,[27] as

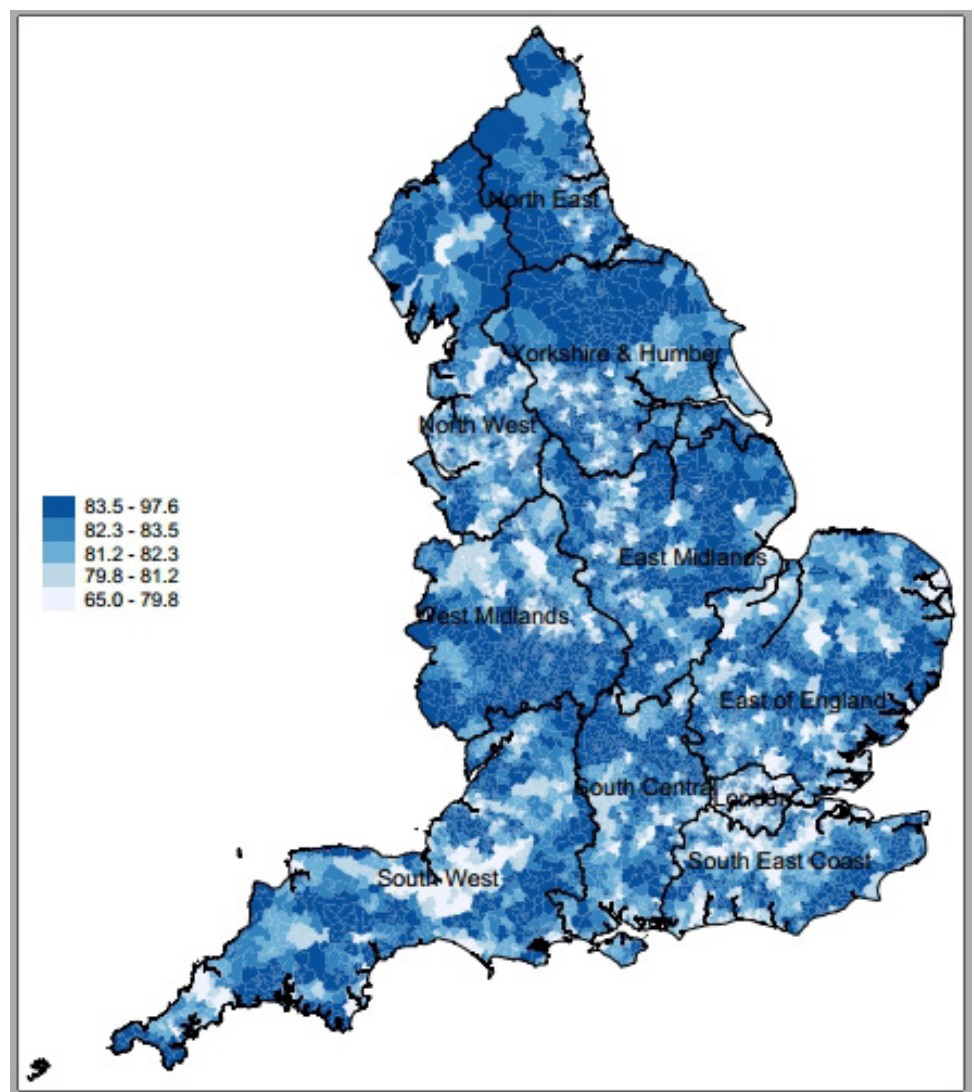

**Figure 3** Population achievement in England for QOF incentivised ACSCs in 2015 (LSOA level). ACSCs, ambulatory care sensitive conditions; LSOA, lower super output area; QOF, Quality and Outcomes Framework.

increased access and quality of primary care, two measures of health system performance, have been associated with lower rates of ACSC admission in many countries.[28] For the UK, the introduction of the QOF was expected to lead to reduced admissions for ACSCs, via increased quality of primary care, especially for incentivised long-term conditions. In this study, we investigated the variation in admissions for incentivised ACSCs between and within regions and we aimed to examine the effects of QOF recorded quality of care and other area and population characteristics on ACSC admissions. We observed wide variations in emergency hospital admissions for incentivised ACSC which implies that they and their associated costs can be reduced, even though it remains uncertain how these can be achieved.[20] Observational evidence on the geographical variation for incentivised ACSCs indicates conditions where interventions to improve care pathways are most needed and identifies areas in the country where ACSC determinants require further investigation.

Furthermore, we observed a very small association between QOF population achievement and ACSC admissions both cross-sectionally and a longitudinally. For example, in 2015/2016 improved QOF population achievement led to a reduction of 0.07% (n=1810) in ACSC admissions. These findings are consistent with previous studies,[19 29] even though the magnitude of the effect varies across studies. These previous studies either evaluated the effects of the QOF on specific conditions[17 30] or were not able to account for census measured covariates such as age and ethnicity.[19 31] However, it is possible that ACSC admissions are influenced by aspects of primary care quality and unmeasured changes that were not captured by the indicators we used and we may have therefore failed to identify larger reductions in ACSC admissions. We may also have failed to identify any large reductions in ACSCs as we compared overall QOF achievement to overall hospitalisation for all incentive conditions. It may be that quality of care for a specific

**Table 1** Area and population characteristics by English region: Quality and Outcomes Framework (QOF) year 12 (2015–2016) census data for 2011: admissions for ACSCs in 2015/2016

| Aggregates | English regions | | | | | | | | | |
|---|---|---|---|---|---|---|---|---|---|---|
| | North East | North West | Yorkshire Humber | East Midlands | West Midlands | East England | London | South West | South Central | South East Coast |
| Population size | 2624621 | 7173835 | 5390576 | 4677038 | 5751000 | 6076451 | 8673313 | 5471180 | 4312297 | 4635616 |
| Admissions for QOF ACSCs | 29354 | 74747 | 49827 | 40621 | 52825 | 49118 | 49738 | 44393 | 25069 | 30540 |
| Rates of admissions for QOF ACSCs per 100000 | 1118.41 | 1041.94 | 924.34 | 868.52 | 918.54 | 808.33 | 573.43 | 811.40 | 581.33 | 658.81 |
| No of practices | 381 | 1165 | 746 | 595 | 892 | 748 | 1369 | 679 | 462 | 575 |
| **Census information (mean across LSOAs)** | | | | | | | | | | |
| IMD 2015 | 26.8 | 26.8 | 25.5 | 20.8 | 24.8 | 17.35 | 23.6 | 18.3 | 14.0 | 16.0 |
| P10 | 6.8 | 6.6 | 6.5 | 5.8 | 6.7 | 4.9 | 8.1 | 5.9 | 3.3 | 4.2 |
| P25 | 1.7 | 11.0 | 11.0 | 9.3 | 11.4 | 8.4 | 13.3 | 9.3 | 5.7 | 7.3 |
| P75 | 38.2 | 39.5 | 36.9 | 29.0 | 36.0 | 23.0 | 32.7 | 23.4 | 19.3 | 20.9 |
| P90 | 52.9 | 55.9 | 54.3 | 42.1 | 50.5 | 34.0 | 41.0 | 35.0 | 29.2 | 31.6 |
| Ethnicity white (%) | 95.6 | 91.0 | 89.6 | 89.6 | 83.6 | 91.0 | 60.6 | 95.4 | 88.9 | 92.7 |
| **QOF information: medians across practice hubs** | | | | | | | | | | |
| List size | 6136 | 5676 | 6695 | 7319 | 5588 | 7612 | 6128 | 7601 | 9300 | 7713 |
| **QOF information (spatially estimated): medians across 2011 LSOAs** | | | | | | | | | | |
| Population achieve (%) | 82.4 | 82.1 | 82.3 | 82.1 | 82.0 | 81.9 | 80.6 | 81.6 | 81.8 | 81.4 |
| P10 | 80.1 | 79.4 | 78.9 | 79.5 | 79.1 | 78.8 | 77.6 | 78.2 | 79.3 | 78.2 |
| P25 | 81.3 | 80.6 | 80.6 | 80.8 | 80.5 | 80.4 | 78.9 | 79.9 | 80.5 | 79.9 |
| P75 | 83.4 | 83.1 | 83.5 | 83.6 | 83.5 | 83.4 | 82.1 | 83.2 | 83.4 | 82.6 |
| P90 | 84.3 | 84.1 | 84.9 | 85.1 | 84.7 | 84.6 | 83.5 | 84.7 | 84.7 | 83.9 |
| **Other Information Across LSOAs** | | | | | | | | | | |
| Rural (%) | 17.5 | 9.8 | 16.5 | 25.4 | 14.8 | 28.1 | 0.1 | 20.1 | 19.7 | 30.2 |

P10, P25, P75, P90 are the respective percentiles for each variable.
ACSCs, ambulatory care sensitive conditions; IMD, Index of Multiple Deprivation; LSOAs, lower super output areas.

**Table 2** Effects of QOF overall population achievement on hospital admissions for QOF incentivised ACSCs

| Year | Negative binomial model | Negative binomial model (over time) | Negative binomial model W/interaction effect for region) | Negative binomial model W/interaction effect for deprivation) |
|---|---|---|---|---|
| **% population achievement** | 0.993 (0.990 to 0.995), <0.001 (0.001) | 0.998 (0.997 to 0.999), <0.001 (0.0005) | 0.989 (0.987 to 0.992), <0.001 (0.001) | 0.998 (0.997 to 0.999), <0.001 (0.0005) |
| Female | 0.852 (0.845 to 0.860), <0.001 (0.003) | 0.810 (0.807 to 0.813) <0.001 (0.001) | 0.852 (0.844 to 0.859), <0.001 (0.002) | 0.810 (0.807 to 0.813) <0.001 (0.001) |
| Index of Multiple Deprivation 2015 | 1.021 (1.020 to 1.021), <0.001 (0.0001) | 1.021 (1.020 to 1.021), <0.001 (0.0001) | 1.023 (1.022 to 1.024), <0.001 (0.0001) | 1.021 (1.020 to 1.021), <0.001 (0.0001) |
| Rural (vs urban) | 0.875 (0.862 to 0.887), <0.001 (0.006) | 0.865 (0.857 to 0.873), <0.001 (0.004) | 0.879 (0.866 to 0.891), <0.001 (0.005) | 0.865 (0.857 to 0.873), <0.001 (0.004) |
| Ethnicity (% white) | 1.0004 (1.0001 to 1.0008), <0.002 (0.005) | 0.999 (0.998 to 0.999), <0.001 (0.0001) | 0.998 (0.998 to 0.999), <0.001 (0.001) | 0.999 (0.998 to 0.999), <0.001 (0.0001) |
| Age (0–04) | Reference category | Reference category | Reference category | Reference category |
| Age (05–09) | 0.988 (0.952 to 1.026), <0.543 (0.019) | 0.792 (0.782 to 0.803), <0.001 (0.005) | 0.988 (0.952 to 1.025), <0.537 (0.020) | 0.792 (0.782 to 0.803), <0.001 (0.005) |
| Age (10–14) | 0.720 (0.689 to 0.752), <0.001 (0.016) | 0.566 (0.556 to 0.576), <0.001 (0.005) | 0.720 (0.689 to 0.752), <0.001 (0.018) | 0.566 (0.556 to 0.576), <0.001 (0.005) |
| Age (15–19) | 0.653 (0.625 to 0.682), <0.001 (0.015) | 0.520 (0.511 to 0.530), <0.001 (0.005) | 0.652 (0.624 to 0.681), <0.001 (0.018) | 0.521 (0.511 to 0.530), <0.001 (0.005) |
| Age (20–24) | )0.688 (0.658 to 0.718) <0.001 (0.015) | 0.488 (0.479 to 0.497) <0.001 (0.005) | 0.687 (0.658 to 0.718) <0.001 (0.018) | 0.489 (0.479 to 0.498) <0.001 (0.005) |
| Age (25–29) | 0.553 (0.529 to 0.577), <0.001 (0.012) | 0.409 (0.402 to 0.417), <0.001 (0.004) | 0.554 (0.530 to 0.576), <0.001 (0.016) | 0.409 (0.402 to 0.417), <0.001 (0.004) |
| Age (30–34) | 0.583 (0.558 to 0.609) <0.001 (0.013) | 0.438 (0.430 to 0.446), <0.001 (0.004) | 0.585 (0.560 to 0.611) <0.015 (0.018) | 0.438 (0.430 to 0.446), <0.001 (0.004) |
| Age (35–39) | 0.665 (0.639 to 0.693) <0.001 (0.013) | 0.531 (0.522 to 0.540), <0.001 (0.004) | 0.667 (0.640 to 0.694) <0.001 (0.020) | 0.531 (0.522 to 0.540), <0.001 (0.005) |
| Age (40–44) | 0.914 (0.880 to 0.949), <0.001 (0.018) | 0.719 (0.708 to 0.730), <0.001 (0.006) | 0.915 (0.881 to 0.950), <0.001 (0.026) | 0.719 (0.708 to 0.730), <0.001 (0.006) |
| Age (45–49) | 1.265 (1.222 to 1.309), <0.001 (0.22) | 1.000 (0.986 to 1.014), <0.972 (0.007) | 1.266 (1.223 to 1.311), <0.001 (0.034) | 1.000 (0.986 to 1.014), <0.974 (0.007) |
| Age (50–54) | 1.759 (1.701 to 1.817), <0.001 (0.030) | 1.388 (1.370 to 1.407), <0.001 (0.010) | 1.760 (1.703 to 1.819), <0.001 (0.047) | 1.388 (1.370 to 1.407), <0.001 (0.010) |
| Age (55–59) | 2.467 (2.388 to 2.550), <0.001 (0.041) | 1.909 (1.884 to 1.934), <0.001 (0.013) | 2.466 (2.386 to 2.548), <0.001 (0.064) | 1.909 (1.884 to 1.934), <0.001 (0.013) |
| Age (60–64) | 3.414 (3.306 to 3.525), <0.001 (0.056) | 2.671 (2.637 to 2.706), <0.001 (0.017) | 3.410 (3.303 to 3.521), <0.001 (0.086) | 2.671 (2.637 to 2.706), <0.001 (0.017) |

Continued

**Table 2** Continued

| Year | Negative binomial model | Negative binomial model (over time) | Negative binomial model W/interaction effect for region) | Negative binomial model W/interaction effect for deprivation) |
|---|---|---|---|---|
| Age (65–69) | 4.590 (4.450 to 4.733), <0.001 (0.072) | 3.838 (3.791 to 3.886), <0.001 (0.024) | 4.585 (4.445 to 4.729), <0.001 (0.112) | 3.838 (3.791 to 3.886), <0.001 (0.024) |
| Age (70–74) | 6.942 (6.734 to 7.157), <0.001 (0.109) | 5.667 (5.600 to 5.736), <0.001 (0.035) | 6.938 (6.729 to 7.153), <0.001 (0.160) | 5.667 (5.599 to 5.735), <0.001 (0.035) |
| Age (75–79) | 9.808 (9.516 to 10.108), <0.001 (0.151) | 7.873 (7.782 to 7.965), <0.001 (0.047) | 9.795 (9.503 to 10.095), <0.001 (0.216) | 7.871 (7.780 to 7.964), <0.001 (0.047) |
| Age (80–84) | 13.380 (12.985 to 13.787), <0.001 (0.205) | 10.714 (10.591 to 10.839), <0.001 (0.063) | 13.368 (12.973 to 13.775), <0.001 (0.285) | 10.712 (10.589 to 10.837), <0.001 (0.063) |
| Age (85+) | 18.448 (17.909 to 19.003) <0.001 (0.279) | 14.363 (14.198 to 14.529), <0.001 (0.084) | 18.445 (17.905 to 19.001) <0.001 (0.381) | 14.359 (14.195 to 14.526), <0.001 (0.084) |
| Imd2015#North East | – | – | Reference Category | – |
| Imd2015#North West | – | – | 0.999 (0.998 to 0.999) <0.002 (0.0004) | – |
| Imd2015#Yorkshire and Humber | – | – | 0.996 (0.995 to 0.997) <0.001 (0.0004) | – |
| Imd2015#East Midlands | – | – | 0.997 (0.996 to 0.998) <0.001 (0.0004) | – |
| Imd2015#West Midlands | – | – | 0.997 (0.996 to 0.997) <0.011 (0.0004) | – |
| Imd2015#East England | – | – | 0.997 (0.996 to 0.996) <0.001 (0.0005) | – |
| Imd2015#London | – | – | 0.990 (0.989 to 0.990) <0.001 (0.0005) | – |
| Imd2015#South East | – | – | 0.992 (0.991 to 0.993) <0.001 (0.0005) | – |
| Imd2015#South East Coast | – | – | 0.992 (0.990 to 0.994) <0.001 (0.0007) | – |
| Imd2015#South West | – | – | 0.994 (0.993 to 0.996) <0.001 (0.0004) | – |
| Imd2015#PA.oval | – | – | | 1.000 (1.000 to 1.000) <0.001 (0.001) |
| Model intercept | 0.0034 (0.0029 to 0.0040), <0.001 (0.0001) | 0.004 (0.003 to 0.004), <0.001 (0.0001) | 0.0065 (0.0055 to 0.0077) <0.001 (0.0001) | 0.0012 (0.0090 to 0.0171) <0.001 (0.0002) |

95% CIs are in brackets, results are reported as incidence rate ratios followed by p values and SEs in parentheses.
ACSCs, ambulatory care sensitive conditions; QOF, Quality and Outcomes Framework.

condition has a greater effect on admissions for that condition, or it may be that quality of care for a specific condition moderates a large effect of quality of care on admissions for other conditions. Despite these issues, our findings are important for policy, due to the increasing proportion of healthcare costs attributed to admissions for ACSCs.[6]

Our results also indicate that admissions for incentivised ACSCs are higher in areas with higher socioeconomic deprivation. The strong relationship between levels of deprivation and admissions for incentivised ACSCs implies that area-level socioeconomic factors, rather than variations in quality of primary care, are more important in predicting risk for ACSC admissions. For example, we found that one additional percentage point in population deprivation is associated with a 2.0% (n=5, 370) increase in admissions for incentivised ACSCs nationally in 2015/2016. A change in IMD score from the

25th centile (9.57) to the median (17.07) would correspond to a 9.75% increase in ACSC admissions (n=26, 180). The extent of this relationship was also highly consistent across the sensitivity analyses reported in the online supplementary appendix. This finding agrees with previous evidence on the association between ACSCs and deprivation[2 19] and suggests that community health-care needs for people living in these areas may not be adequately met, and this may be contributing to widening health inequalities. Furthermore, patients from deprived areas are known to attend ED more frequently and for less serious conditions, while they access outpatient care mostly via emergency channels.[32] This may also partially explain the increased effect of deprivation on admissions for incentivised ACSCs.

We found that ACSC hospital admission rates were higher for men, and—as expected—increased with age. Urbanicity was also associated with increased risk for incentivised ACSC admission, and this may indicate the increased likelihood of hospitalisation in urban areas and metropolitan areas where geographical access is easier.

## CONCLUSION

Our findings find wide within and between regional differences in admissions for incentivised ACSCs. This is a particular concern for NHS, because hospital admissions for these conditions are increasing and are potentially avoidable. Our findings also indicate that financial incentives to improve the quality of care and reduce variations in practice performance are associated with very small reductions in hospital admissions for ACSCs. This suggests that there are limits to the impact financial incentive schemes in primary care can have on avoidable admissions. Finally, our findings pertain to other countries that incentivise similar processes of care, however, different systems may respond differently and our results should be interpreted with caution.

**Author affiliations**
[1]NIHR School for Primary Care Research, Centre for Primary Care, Division of Population Health, Health Services Research and Primary Care, University of Manchester, Manchester, UK
[2]Manchester Academic Health Sciences Centre (MAHSC), Manchester, UK
[3]Health Organisation, Policy and Economics, Centre for Primary Care, Division of Population Health, Health Services Research and Primary Care, University of Manchester, Manchester, UK
[4]Centre for Mental Health and Safety, Division of Psychology & Mental Health, The University of Manchester, Manchester, UK
[5]NIHR Greater Manchester Patient Safety Translational Research Centre, University of Manchester, Manchester, UK
[6]Centre for Suicide Prevention, Division of Psychology and Mental Health, University of Manchester, Manchester, UK
[7]Greater Manchester Mental Health Trust, Manchester, UK
[8]Department of Health Sciences, University of York, York, UK
[9]Centre for Pharmacoepidemiology and Drug Safety, School of Health Sciences, Faculty of Biology, Medicine and Health, University of Manchester, Manchester, UK
[10]Faculty of Biology, Medicine & Health, Division of Informatics, Imaging and Data Sciences, University of Manchester, Manchester, UK

**Acknowledgements** We would like to thank the Office of National Statistics for the wealth of information they have collected and systematically organised, which made this study possible.

**Contributors** EK, CG, DA and LM designed the study. CG extracted the data from all sources, performed the analyses and drafted the manuscript. EK, RW, DA, TD, LM and NK critically revised the manuscript. CG is the guarantor of this work and, as such, had full access to all the data in the study and takes responsibility for the integrity of the data and the accuracy of the data analysis.

**Funding** This study is funded by the National Institute for Health Research School for Primary Care Research (NIHR SPCR), through CG's PhD. This report is independent research by the National Institute for Health Research. LM acknowledges financial support from the MRC Skills Development Fellowship (MR/N015126/1).

**Disclaimer** The views expressed in this publication are those of the authors and necessarily those of the NHS, the National Institute for Health Research, the Department of Health, or the MRC.

**Map disclaimer** The depiction of boundaries on this map does not imply the expression of any opinion whatsoever on the part of BMJ (or any member of its group) concerning the legal status of any country, territory, jurisdiction or area or of its authorities. This map is provided without any warranty of any kind, either express or implied.

**Competing interests** None declared.

**Patient consent for publication** Not required.

**Provenance and peer review** Not commissioned; externally peer reviewed.

**Data availability statement** Data may be obtained from a third party and are not publicly available. The data used in this study are freely available except for the Hospital Episode Statistics data which were obtained under a special license with NHS Digital. The authors are happy to share in an organised and cleaned final dataset all publicly available data.

**ORCID iDs**
Christos Grigoroglou http://orcid.org/0000-0003-1621-8648
Luke Munford http://orcid.org/0000-0003-4540-6744
Evangelos Kontopantelis http://orcid.org/0000-0001-6450-5815

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
