## [Reviewer comments · BMJ Open]

ARTICLE DETAILS

TITLE (PROVISIONAL)	The impact of a national primary care pay-for-performance scheme on ambulatory care sensitive hospital admissions: A small area analysis in England
AUTHORS	Grigoroglou, Christos; Munford, Luke; Webb, Roger; Kapur, Navneet; Doran, Tim; Ashcroft, Darren; Kontopantelis, Evangelos

VERSION 1 – REVIEW

REVIEWER	Louisa Jorm Centre for Big Data Research in Health UNSW Sydney
REVIEW RETURNED	22-Jan-2020

GENERAL COMMENTS	It is not clear exactly what the key exposure variable 'QOF population achievement' is (e.g. completeness of selected measures of the QOF, or specific values of QOF indicators?). The cited table which is meant to detail the QOF indicators used (Supplementary Table S2) is not actually included in the supplementary material, and the reader has no context as to what it is measuring (e.g. what's the numerator, denominator, range of values, etc). For example, the authors cite there was only a small effect of QOF population achievement on rates of hospitalisation, by comparing IRRs from the QOF variable (for a 1% change in 'quality of care') to the IRRs from other variables (e.g. urban vs rural). But what does one unit of change in this QOF variable actually mean - and is this 1% change equivalent to the effect size observed for the other variables it is being compared to? It's impossible to assess the robustness of analysis and validity of the conclusions without sufficient details on the exposure, but highly likely the direct comparison of IRRs are not appropriate and the conclusions are overstated. While the paper cannot be adequately assessed without sufficient detail of the exposure, some further preliminary issues for clarification are below: QOF calculation – why are exceptions (patients deemed ineligible for treatment) included in the denominator of the QOF? (pg 6, line 58) Zero-inflated models are two-part models (i.e. binary and negative binomial outcomes). Which part have the predictor variables been included in, and why? Reporting this model adds little to the paper, and seems better suited to supplementary material. The variables used are often only partially described, and often very confusing. e.g. it would be helpful to describe what 'primary care practice hubs' are in the text – the way these are described (e.g. page 6 line 48) it is not clear if this is the number of 'hubs', people within a 'hub', geographic regions with a hub, etc.
---

REVIEWER	Makoto Kaneko Department of Family and Community Medicine, Hamamatsu University School of Medicine, Japan
REVIEW RETURNED	15-Feb-2020

GENERAL COMMENTS	Thank you for your great work. I enjoyed reading the article. My comments as below are based on the STROBE checklist for observational study (http://www.equator-network.org/reporting-guidelines/strobe/) and I hope these comments improve your article. This is a really well-designed and well-written paper. Although there are several limitations, the authors have already taken them into account. Major concerns #1. The breakdown of the ACSCs such as top-3 ACSCs is useful information for readers in other countries. If you can, please add the overview of the ACSCs in the study as a supplementary file. #2. What is the implication of the study for other countries with different health care system? The information is helpful for international readers and allows the study to become more popular.
---

REVIEWER	Marco Fonzo Department of Cardiological, Thoracic and Vascular Sciences Hygiene and Public Health Unit University of Padova, Italy
REVIEW RETURNED	27-Feb-2020

GENERAL COMMENTS	In general Grigoroglou and colleagues conducted a comprehensive study on QOF incentivized ACSCs in England, assessing the impact not only of the QOF scheme, but also taking into account a number of socio-economic determinants. Using a small area analysis, the work can provide a useful insight at the very local level: the preparation of maps with such a high degree of detail represents a remarkable added value. Introduction The introduction is straight-to-the-point and clearly highlights the need for investigating ACSC. However, some passages are not completely clear. I appreciated the digression on the QOF (p 4, third paragraph) and I found it necessary for international readers who are not necessarily familiar with the English NHS. P 4 row 46: while the expression “incentivised condition” is instantly understandable, the use of “incentivised intervention” sounds a little bit confusing. If I did not see wrong, it is the only time this expression was used. P 5 row 1-9: please further explain this passage. P 5 row 16-24: please clarify this passage, in particular the connection between the main and the subordinate clause. The study aim is overall clear. Next to the spatially description of ACSC, I would consider also the spatially description of QOF achievement (as of Figure 3).
---

	Methods In my opinion, the definition of what is an ACSC is very important, even though it may appear redundant. Supplementary material and ref. 19 provide a clear definition, however it would be interesting to move Table 1 from the appendix to the manuscript (if allowed by editorial guidelines). In addition, I would further stress the fact that QOF addresses only some of ACSC. This manuscript considers only incentivised ACSC, not all ACSC (and this should be highlighted in both text and all tables). I would suggest to provide a more detailed definition of “QOF population achievement”. It was necessary to me to consult an additional reference. Results To improve readability of results, I would add in Table 1 a row showing ACSC per 100,000 people, as reported in p 7 row 49-51. Table 2 looks redundant: keep most valuable findings in it, while moving the rest in the supplementary material. Following the study aim c): to examine how characteristics investigated drive variability across English regions, I guess that Table 2 (supplementary material) is essential. Therefore, I would show those findings in the manuscript. P 8 row 26-30: the study did not really focus on the economic aspect of the matter. However, from time to time there are throughout the document mention of costs which I would delete. Moreover, in the results part, I would stick with findings from the study at issue, nothing else. Discussion/conclusions While the focus of the paper – understandably – was on the effect of QOF, data show a relevant role of socioeconomic determinants, which are not extensively discussed. I believe that the manuscript would benefit from a further description of the context and potential interactions between all socioeconomic determinants investigated. Conclusions. Coherently with the whole work and the magnitude of the effect of QOF, I would say that “financial incentives [...] are associated with relatively moderate reductions...”. Conclusions. I believe that the very last sentence goes beyond the purpose of the study and the capacity of inference from findings. Thus, I would focus more on the associations investigated rather than considering “primary care interventions in general”.
--	---

REVIEWER	Young-Rock Hong University of Florida / USA
REVIEW RETURNED	16-Mar-2020

GENERAL COMMENTS	Overall this is a great work that tackles a question of the impact of the most prominent P4P on avoidable hospital admissions (ACSC related) at population-level. I read this with interest as I am a US-based researcher who is interested in the value-based program (VBP) and its impact on population health. P4P/VBP is a major issue in the financing of health care around the world. Further elucidating the real-world effects of P4P is needed and this study does this job. The authors should be applauded for their great efforts. I would recommend this manuscript for publication with some minor suggestions.  • Introduction (page 4, lines 55-60): If word limit allowing, it would be better to expand further why there is mixed evidence on some
--

	chronic conditions. Given that diabetes or heart failure are classic chronic conditions and in the ACSC category as well, better ambulatory care is supposed to be able to prevent hospitalizations for those conditions.  • Method/Discussion: In the same vein, if it's not too much, I would like to see which specific ACSC was impacted most. The authors mentioned in the introduction that the QOF has resulted in a reduction of admissions for some chronic conditions (e.g., COPD, asthma) but didn't work for heart disease. Discussing whether this difference was still the case or not would make a stronger case for "why this study is important." • Method: the primary outcome was the rate of emergency admissions. I see the model accounted for age; but especially for population-level and area-focused outcomes, they should be presented in age-adjusted given geographic variations (e.g., >32000 English LSOAs). At least, the authors should show there is no difference in crude rate and age-adjusted. But I am still inclined to use the age-standard approach is more representative.
--	---

VERSION 1 – AUTHOR RESPONSE

Reviewer: 1

Reviewer Name: Louisa Jorm

Institution and Country:

Centre for Big Data Research in Health
UNSW Sydney

Please state any competing interests or state 'None declared': None declared

Please leave your comments for the authors below

It is not clear exactly what the key exposure variable 'QOF population achievement' is (e.g. completeness of selected measures of the QOF, or specific values of QOF indicators?). The cited table which is meant to detail the QOF indicators used (Supplementary Table S2) is not actually included in the supplementary material, and the reader has no context as to what it is measuring (e.g. what's the numerator, denominator, range of values, etc). For example, the authors cite there was only a small effect of QOF population achievement on rates of hospitalisation, by comparing IRRs from the QOF variable (for a 1% change in 'quality of care') to the IRRs from other variables (e.g. urban vs rural). But what does one unit of change in this QOF variable actually mean - and is this 1% change equivalent to the effect size observed for the other variables it is being compared to? It's impossible to assess the robustness of analysis and validity of the conclusions without sufficient details on the exposure, but highly likely the direct comparison of IRRs are not appropriate and the conclusions are overstated.

While the paper cannot be adequately assessed without sufficient detail of the exposure, some further preliminary issues for clarification are below:

QOF calculation – why are exceptions (patients deemed ineligible for treatment) included in the denominator of the QOF? (pg 6, line 58)

Zero-inflated models are two-part models (i.e. binary and negative binomial outcomes). Which part have the predictor variables been included in, and why? Reporting this model adds little to the paper, and seems better suited to supplementary material.

The variables used are often only partially described, and often very confusing. e.g. it would be helpful

to describe what 'primary care practice hubs' are in the text – the way these are described (e.g. page 6 line 48) it is not clear if this is the number of 'hubs', people within a 'hub', geographic regions with a hub, etc.

Response: Thank you for your careful suggestions as to how we can improve the manuscript. QOF population achievement essentially captures general practice performance within the QOF. You are right that the definition of QOF population achievement was not clear before and we have now revised the manuscript to describe this measure in more detail.

In p 6 the manuscript now reads “We captured QOF performance by calculating population achievement, defined as $PA_{\text{oval}} = \frac{\sum N_i}{\sum [(D_i + E_i)]}$ (1) where the numerator represents the sum of all patients who have actually received the care (N_i) described in the relevant indicator (indicator i), the denominator represents the sum of the number of patients from the appropriate disease register who are eligible to receive the care described in the relevant QOF indicator (D_i) and the sum of the number of patients who have been exception reported for the relevant QOF indicator (E_i). (2, 3)”. We apologise for the omission of table S2 in the original submission of the supplementary material. We amended the supplementary material to add Table 2 which features detailed information on all QOF numerators used in the composite measure “QOF population achievement”. The table includes information on range of values, years in which the indicators were active, detailed description and indicator name. We hope the table will clarify the reviewer’s question on what exactly QOF population achievement measures.

Regarding IRR comparisons, we agree with the reviewer that in our original submission some parts were perhaps confusing. We have made changes in the manuscript to clarify how we interpret the magnitude of effects. In p.11 par.2 it now reads “Furthermore, we observed a very small association between QOF performance and ACSC admissions both cross-sectionally and a longitudinally. For example, in 2015/16 improved QOF performance led to a reduction of 0.07% ($n=1810$) in ACSC admissions”. Our aim was not to directly compare IRRs as this would require a formal statistical test to justify any significant comparison between IRRs. In our study, we compare absolute rather than relative effects. For example, an IRR of 0.993 amounts to $n=1810$ fewer admissions for incentivised ACSCs in 2015/16. Similarly a one point increase in deprivation (IRR = 1.020) amounts to $n=5,370$ more admissions for incentivised ACSCs. Moreover, rural areas will have on average 32,313 less admissions for incentivised ACSCs in 2015/16 (IRR=0.875) when compared to urban areas. Given that most practices already score very high on QOF recorded performance and levels of performance change only a little, year by year, we argue that the QOF can only have a minimal impact on hospital admission for incentivised ACSCs.

In terms of the exception reporting, NHS Digital provides guidelines for the measurement of QOF achievement which we incorporated in our measure of population achievement. There are currently two ways to measure QOF achievement, one with exception included in the denominator and one without. In our study, we opted for the first option for the following reasons. QOF financial payments to practices are typically based on the proportion of patients for which targets are met between a lower achievement threshold and an upper threshold that varies according to the relevant indicator. The upper payment threshold for the period of analyses was between 50 and 90 percent for quality indicators. According to this, practices can earn the maximum points available for an indicator without necessarily achieving the targets for all relevant patients. For example, a practice can deliver the required care to fewer than 100 per cent of its patients (often around 90 per cent) to achieve the full (100 per cent) points available by excluding patients for specific indicators (exception reporting). For this reason, there is an important distinction between the underlying achievement (net of exceptions) and percentage achievement in terms of QOF points available for specific indicators. Underlying achievement (net of exceptions) does not account for all patients covered by an indicator, as it takes no account of “exceptions” (patients to whom the indicator applies, but who are not included in the indicator denominator according to agreed exception criteria) and does not reflect any improvements

in quality above the upper threshold. Percentage of patients receiving the intervention gives a more accurate indication of the rate of the provision of an intervention as the denominator for this measure covers all patients to whom the indicator applies, regardless of exception status (i.e. indicator exceptions and indicator denominator). (4)(ref). In our study we were interested in assessing the provision of good quality of Primary Care for the whole of the English population and including exception gives us a more accurate picture of the rate of provision for the most relevant population.

Finally, Primary Care “hubs” are defined as geographic areas (i.e. LSOAs) that contain one or more general practices. We feel that this was not clear before in the manuscript and we have rephrased to clarify this distinction. In the manuscript, p.6 last par. it now reads “In England, general practice healthcare coverage is comprehensive as 95% of the UK population is estimated to be registered with a practice, and over 99% of registered patients attend practices participating in the QOF. (5) LSOA general practice performance data were obtained from the practice-level data in two ways. First, for those localities (i.e. LSOAs) containing at least one general practice, practice-level information were aggregated at the LSOA level and this allowed us to account for multiple practices in an LSOA. These primary care practice ‘hubs’ varied between 6,455 in the 3rd year of the QOF (2006/07) to 5,736 in the year of our analysis (2015/16). We hope this clarifies the reviewer’s question.

Reviewer: 2

Reviewer Name: Makoto Kaneko

Institution and Country: Department of Family and Community Medicine, Hamamatsu University School of Medicine, Japan

Please state any competing interests or state ‘None declared’: None declared

Please leave your comments for the authors below

Thank you for your great work.

I enjoyed reading the article.

My comments as below are based on the STROBE checklist for observational study (<http://www.equator-network.org/reporting-guidelines/strobe/>)

and I hope these comments improve your article.

This is a really well-designed and well-written paper. Although there are several limitations, the authors have already taken them into account.

Major concerns

#1. The breakdown of the ACSCs such as top-3 ACSCs is useful information for readers in other countries. If you can, please add the overview of the ACSCs in the study as a supplementary file.

Response: Thank you this is a very good point. In the supplementary file, we added a figure to provide an overview of all incentivised ACSCs over our study period. Our findings indicate that admissions for incentivised ACSCs were the highest for COPD, stroke and CHD. We have also added a sentence in the manuscript in p.8 par.1 where it now reads “There were 268, 509 ACSC admissions with a primary diagnosis linked to QOF incentivised conditions in 2015/16 alone while admissions for heart disease and COPD accounted for the majority of admissions for incentivised ACSCs over time (Figure 1 in supplementary material).”

#2. What is the implication of the study for other countries with different health care system? The

information is helpful for international readers and allows the study to become more popular.

Response: Thank you for your suggestion and we think it is valuable to know whether similar schemes can be effective in other countries, however, we are not in a position to ascertain the implications of our study for other countries. We are confident that our findings pertain to other countries that have implemented or are on the process of implementing similar national interventions with similar processes of care, however each health care system is unique and direct comparisons should be made with caution. Nevertheless, we agree that a statement within this context would be useful for international readers. In the conclusion of the manuscript it now reads “Finally, our findings pertain to other countries that incentivise similar processes of care, however, different systems may respond differently and our results should be interpreted with caution.”

Reviewer: 3

Reviewer Name: Marco Fonzo

Institution and Country:

Department of Cardiological, Thoracic and Vascular Sciences

Hygiene and Public Health Unit

University of Padova, Italy

Please state any competing interests or state 'None declared': None declared.

Please leave your comments for the authors below

In general

Grigoroglou and colleagues conducted a comprehensive study on QOF incentivized ACSCs in England, assessing the impact not only of the QOF scheme, but also taking into account a number of socio-economic determinants. Using a small area analysis, the work can provide a useful insight at the very local level: the preparation of maps with such a high degree of detail represents a remarkable added value.

Response: Thank you. We appreciate your positive view of our work.

Introduction

The introduction is straight-to-the-point and clearly highlights the need for investigating ACSC. However, some passages are not completely clear. I appreciated the digression on the QOF (p 4, third paragraph) and I found it necessary for international readers who are not necessarily familiar with the English NHS.

P 4 row 46: while the expression “incentivised condition” is instantly understandable, the use of “incentivised intervention” sounds a little bit confusing. If I did not see wrong, it is the only time this expression was used.

Response: We have now changed the term “incentivised intervention” to “incentivised processes of care”.

P 5 row 1-9: please further explain this passage.

Response: We understand that this passage was not clear before and we rephrased. The manuscript now reads “Previous evidence suggested that the scheme may have reduced ACSC admissions for conditions that were included in the scheme relative to ACSC admissions that were not included in the scheme - by 3% in 2004/2005 and 8% in 2010/11 - and this was mainly driven by relative

reductions in emergency admissions for coronary heart disease.(6)”

P 5 row 16-24: please clarify this passage, in particular the connection between the main and the subordinate clause.

Response: We agree with the reviewer that this statement was not clear before. The main and subordinate causes relate to the same context and we rephrased to reflect that. The passage now reads “These variations have also revealed that hospital admissions for ACSC are concentrated in areas of high socioeconomic deprivation.(7)”

The study aim is overall clear. Next to the spatially description of ACSC, I would consider also the spatially description of QOF achievement (as of Figure 3).

Response: We added the aim of spatially describing QOF achievement (Figure 3) in our study aims.

Methods

In my opinion, the definition of what is an ACSC is very important, even though it may appear redundant. Supplementary material and ref. 19 provide a clear definition, however it would be interesting to move Table 1 from the appendix to the manuscript (if allowed by editorial guidelines).

Response: We added the following statement to clarify the set of ACSCs that were used in our analysis. In p.6 it now reads “To measure ACSCs we followed the work by Harrison et al. (6); 8 specific conditions were chosen based on the list of ACSCs which are used for measuring system performance in the NHS (8) and we also included complications of diabetes associated with hypoglycaemia. (9) The 8 QOF incentivised conditions included: asthma, coronary heart disease, chronic obstructive pulmonary disease, diabetes (all admissions for hyperglycaemia and hypoglycaemia), epilepsy, heart failure, hypertension and stroke.” We initially considered moving table 1 to the main text, however, in the above statement we specify which conditions were investigated and therefore table 1 which specifies ICD-10 codes rather than conditions was considered redundant.

In addition, I would further stress the fact that QOF addresses only some of ACSC. This manuscript considers only incentivised ACSC, not all ACSC (and this should be highlighted in both text and all tables).

Response: The amended manuscript and tables refer now to “incentivised ACSCs”

I would suggest to provide a more detailed definition of “QOF population achievement”. It was necessary to me to consult an additional reference.

Response: Thank you. This point was raised also by reviewer 1, therefore we now provide a more clear definition of “QOF population achievement” in p.7.

Results

To improve readability of results, I would add in Table 1 a row showing ACSC per 100,000 people, as reported in p 7 row 49-51.

Response: This is a valid point. However, we feel that it would be redundant for the manuscript to report both crude numbers and rates per 100,000 people and we therefore opted to report total number of cases to describe the total burden of admissions for incentivised ACSCs in England. In p7, the reporting of ACSCs per 100,000 people was based on Figure 2 (i.e. spatial map of ACSCs) where we necessarily used the adjusted rate due to data sensitivity issues at the LSOA level.

Table 2 looks redundant: keep most valuable findings in it, while moving the rest in the supplementary material.

Response: We have now amended table 2 to include only the most valuable findings as well as some sensitivity analysis results that you requested in your next point.

Following the study aim c): to examine how characteristics investigated drive variability across English regions, I guess that Table 2 (supplementary material) is essential. Therefore, I would show those findings in the manuscript.

Response: See response above.

P 8 row 26-30: the study did not really focus on the economic aspect of the matter. However, from time to time there are throughout the document mention of costs which I would delete. Moreover, in the results part, I would stick with findings from the study at issue, nothing else.

Response: We agree with the reviewer that we did not investigate economic aspects of the QOF and we have therefore removed most of the statements referring to costs. We maintained some instances referring to costs in order to provide information on why preventing admissions for ACSCs can be a powerful tool to minimise costs in the NHS.

Discussion/conclusions

While the focus of the paper – understandably – was on the effect of QOF, data show a relevant role of socioeconomic determinants, which are not extensively discussed. I believe that the manuscript would benefit from a further description of the context and potential interactions between all socioeconomic determinants investigated.

Response: We agree with the reviewer that a further description on the context of socioeconomic deprivation is needed. We added the following statement in p.12 par.1 Furthermore, patients from deprived areas are known to attend emergency department (ED) more frequently and for less serious conditions, while they access outpatient care mostly via emergency channels. (10) This may also partially explain the increased effect of deprivation on admissions for incentivised ACSCs.

Conclusions. Coherently with the whole work and the magnitude of the effect of QOF, I would say that “financial incentives [...] are associated with relatively moderate reductions...”.

Response: Thank you for this point, however we feel that the observed effect were minimal when compared to previous studies. Our results indicated that QOF recorded performance accounted for 1,870 fewer admissions for incentivised ACSCs in 2015/16. We feel that this is a very small number especially when compared to previous studies, which found moderate effects of the QOF but did not control for area and population characteristics. For example the study by Harisson et al. found that the QOF led to a reduction of approximately 53,000 hospital admissions in England in 2010/11 and this effect was found for exactly the same incentivised ACSCs that we used in our paper. Our findings indicate a much lower effect of the QOF on incentivised ACSCs.

Conclusions. I believe that the very last sentence goes beyond the purpose of the study and the capacity of inference from findings. Thus, I would focus more on the associations investigated rather than considering “primary care interventions in general”.

Response: We have now amended this statement to remove the term “primary care interventions in general”.

Reviewer: 4

Reviewer Name: Young-Rock Hong

Institution and Country: University of Florida / USA

Please state any competing interests or state 'None declared': None declared

Please leave your comments for the authors below

Overall this is a great work that tackles a question of the impact of the most prominent P4P on avoidable hospital admissions (ACSC related) at population-level. I read this with interest as I am a US-based researcher who is interested in the value-based program (VBP) and its impact on population health. P4P/VBP is a major issue in the financing of health care around the world. Further elucidating the real-world effects of P4P is needed and this study does this job. The authors should be applauded for their great efforts. I would recommend this manuscript for publication with some minor suggestions.

- Introduction (page 4, lines 55-60): If word limit allowing, it would be better to expand further why there is mixed evidence on some chronic conditions. Given that diabetes or heart failure are classic chronic conditions and in the ACSC category as well, better ambulatory care is supposed to be able to prevent hospitalizations for those conditions.

Response: We agree with the reviewer that a further understanding of the mixed evidence for different chronic conditions is needed. However, we feel that this is not related to our primary research question; that is the assessment of the QOF as a country-wide intervention aimed to improve the overall quality of Primary Care in England and its association with hospital admissions for incentivised ACSCs. It is possible that different indicators for different conditions work better than others and these are implication that we would not be able to discuss in detail within the present study. It is likely that analyses of individual conditions should be expanded across several pieces of work and this is something we aim to return to in the future.

- Method/Discussion: In the same vein, if it's not too much, I would like to see which specific ACSC was impacted most. The authors mentioned in the introduction that the QOF has resulted in a reduction of admissions for some chronic conditions (e.g., COPD, asthma) but didn't work for heart disease. Discussing whether this difference was still the case or not would make a stronger case for "why this study is important."

Response: Similarly to our previous response we feel that this question, although very important and clinically relevant, is not related to our primary research question for the reasons described before. If it helps, we have provided a table in the supplementary material that describes the breakdown of admissions for incentivised ACSCs over our study period as requested by reviewer 2.

- Method: the primary outcome was the rate of emergency admissions. I see the model accounted for age; but especially for population-level and area-focused outcomes, they should be presented in age-adjusted given geographic variations (e.g., >32000 English LSOAs). At least, the authors should show there is no difference in crude rate and age-adjusted. But I am still inclined to use the age-standard approach is more representative.

Response: This is a good point and a reasonable alternative. As the reviewer can appreciate, the choice is rather subjective and each approach has its own advantages and disadvantages. We opted for our current approach, which we deemed more suitable to describe the total burden of ACSC admissions for incentivised conditions in each locality. Nevertheless, if the reviewer would prefer to see admissions for incentivised ACSCs expressed as age adjusted rates we are happy to add that information.

References:

1. Kontopantelis E, Springate DA, Ashworth M, Webb RT, Buchan IE, Doran T. Investigating the relationship between quality of primary care and premature mortality in England: a spatial whole-population study. *Bmj*. 2015;350:h904.
2. Kontopantelis E, Springate D, Reeves D, Ashcroft DM, Valderas JM, Doran T. Withdrawing performance indicators: retrospective analysis of general practice performance under UK Quality and Outcomes Framework. *BMJ*. 2014;348:g330.
3. Kontopantelis E, Buchan I, Reeves D, Checkland K, Doran T. Relationship between quality of care and choice of clinical computing system: retrospective analysis of family practice performance under the UK's quality and outcomes framework (vol 3, e003190, 2013). *Bmj Open*. 2013;3(8).
4. Digital N. Quality and Outcomes Framework, Achievement, prevalence and exceptions data 2018-19: Technical Annex. 2018.
5. Digital. N. Quality and Outcomes Framework Achievement, Prevalence and Exceptions Data, 2015/16: Frequently Asked Questions. . NHS Digital. 2016.
6. Harrison MJ, Dusheiko M, Sutton M, Gravelle H, Doran T, Roland M. Effect of a national primary care pay for performance scheme on emergency hospital admissions for ambulatory care sensitive conditions: controlled longitudinal study. *Bmj*. 2014;349:g6423.
7. Tian Y, Dixon A, Gao H. Emergency hospital admissions for ambulatory care-sensitive conditions: identifying the potential for reductions. *The King's Fund*. 2012:1-13.
8. Purdy S, Griffin T, Salisbury C, Sharp D. Ambulatory care sensitive conditions: terminology and disease coding need to be more specific to aid policy makers and clinicians. *Public health*. 2009;123(2):169-73.
9. Dusheiko M, Doran T, Gravelle H, Fullwood C, Roland M. Does higher quality of diabetes management in family practice reduce unplanned hospital admissions? *Health services research*. 2011;46(1 Pt 1):27-46.
10. McCormick B, Hill P-S. Are hospital services used differently in deprived areas? Evidence to identify commissioning challenges.

VERSION 2 – REVIEW

REVIEWER	Makoto Kaneko Hamamatsu University School of Medicine, Japan
REVIEW RETURNED	15-Apr-2020
GENERAL COMMENTS	Thank you for revising the manuscript. I have no further comments.
REVIEWER	Young-Rock Hong University of Florida / USA
REVIEW RETURNED	21-Apr-2020
GENERAL COMMENTS	Overall, the authors were thorough and thoughtful in their responses to each reviewer's queries and requests, though they seemed somewhat reluctant to address my suggestions. Given that my things are minor, if the Editors also resonate with me, I would support a publication of this manuscript.